🔓 | **Open Peer Review** | Clinical Microbiology | Observation

# Clinical outcomes of phenotype-guided treatment in group D carbapenemase-producing *Enterobacterales* bacteremia

Jinghao Nicholas Ngiam,[1] Matthew Chung Yi Koh,[1] Nicholas Jian Hao Chan,[1] Jeanette Teo,[2] Ka Lip Chew[2]

**ABSTRACT** International guidance recommends ceftazidime-avibactam for OXA-48-type carbapenemase-producing *Enterobacterales* (OXA-48-CPE) infections. However, OXA-48-CPE lacking extended spectrum and AmpC β-lactamases may remain susceptible to third-generation cephalosporins (3GC), potentially allowing treatment without avibactam. OXA-23 also has variable hydrolytic activity against 3GC. This study reviewed the outcomes of OXA-CPE bloodstream infections treated based on phenotypic susceptibility testing. A retrospective review of OXA-48-CPE and OXA-23-CPE bloodstream infections was conducted from February 2022 to July 2024. Data on antimicrobial susceptibility, patient background, infection source, treatments, and outcomes were analyzed. Whole-genome sequencing (WGS) determined the presence of β-lactamase genes in available isolates. Ten bloodstream infection episodes occurred in nine patients (eight OXA-48-like *Escherichia coli* [$n = 4$], *Klebsiella pneumoniae* [$n = 4$], and one OXA-23-CPE *Proteus mirabilis*), with a urinary source in 44.4% ($n = 4$). Of the isolates, 5/9 (55.5%) were 3GC-susceptible. WGS did not identify any extended-spectrum β-lactamases in these isolates. Three 3GC-susceptible cases (two OXA-48 and one OXA-23) were successfully treated with ceftriaxone. Among the 3GC-resistant cases, two were treated with ceftazidime-avibactam, and two with high-dose meropenem alone or in combination with aztreonam. One recurrence was observed in a case treated with ceftazidime-avibactam. Mortality was low, with one death reported in a patient treated with ceftazidime-avibactam. In this small study, 55.5% of OXA-48-CPE and OXA-23-CPE isolates retained 3GC susceptibility. When used for therapy, 3GC and non-β-lactam antibiotics demonstrated clinical efficacy where antibiotic susceptibility was demonstrated.

**IMPORTANCE** The Infectious Diseases Society of America guidance document recommends the use of ceftazidime-avibactam for the treatment of OXA-48-type carbapenemase-producing *Enterobacterales* (OXA-48-CPE). However, this antibiotic is expensive and not always available in some settings. The clinical outcomes of OXA-48-CPE and OXA-23-CPE bloodstream infections when treated with third-generation cephalosporin (3GC) remain unclear. Among 10 episodes in nine patients, we found that five isolates (55.5%) were 3GC-susceptible, with three cases successfully treated with ceftriaxone. Whole-genome sequencing demonstrated the absence of extended-spectrum β-lactamases in the 3GC-susceptible isolates, which aligns with phenotypic 3GC susceptibility. Mortality was low (1/9). Many OXA-48-CPE and OXA-23-CPE infections retain 3GC susceptibility, raising the possibility that these agents may be viable alternatives to ceftazidime-avibactam in select cases.

**KEYWORDS** OXA-48, carbapenemase, carbapenem-resistant, *Enterobacterales*, outcomes, antibiotic choice

**Peer Reviewer** Yehuda Carmeli, Israel Ministry of Health, Tel-Aviv, Israel

Address correspondence to Ka Lip Chew, ka_lip_chew@nuhs.edu.sg, or Matthew Chung Yi Koh, matthew.koh@mohh.com.sg.

Jinghao Nicholas Ngiam and Matthew Chung Yi Koh contributed equally to this article. The author order was determined by seniority.

The authors declare no conflict of interest.

Carbapenemases continue to drive increasing resistance of *Enterobacterales* to carbapenems and are classified into class A, B (metallo-β-lactamase), and D based on the Ambler classification. Class D carbapenemases, also known as oxacillinase, are further separated into two phylogenetic groups, with OXA-23 being the predominant one in Group I, while Group II consists of variants of OXA-48, typically described as OXA-48-like carbapenemases (1). Despite both being oxacillinases, OXA-23 and OXA-48 are quite distinct epidemiologically.

OXA-23 is more often chromosomal and is the most commonly associated mechanism of carbapenem resistance in *Acinetobacter baumannii*. In *Enterobacterales* (2), OXA-23 is conversely an uncommon mechanism of carbapenem resistance, mostly seen in a single lineage of *Proteus mirabilis* that has disseminated globally (3, 4). Similar to *A. baumannii*, OXA-23 in *P. mirabilis* is also more commonly chromosomally encoded, which may explain the limited horizontal transmission to other species seen. OXA-48-like carbapenemase-producing *Enterobacterales* (OXA-48-CPE), however, are usually plasmid-mediated and have disseminated globally, being one of the three most commonly reported carbapenemases in CPE. They were first reported in Singapore in 2014 and have since been on the rise, causing outbreaks in nosocomial settings (5–7).

Despite differences between OXA-48 and OXA-23 (referred to collectively as OXA), they do share some similarities in the context of *Enterobacterales*. These enzymes demonstrate the ability to hydrolyze carbapenems, but may lack intrinsic activity or demonstrate variable activity against the third-generation cephalosporins (3GC) (5, 8, 9). However, organisms that co-harbor other resistance mechanisms, such as extended-spectrum β-lactamases (ESBLs) or AmpC enzymes, will demonstrate phenotypic resistance to 3GCs (8, 10). In addition, the efficiency with which OXA carbapenemase hydrolyzes carbapenems is also variable. Many OXA-CPE may remain phenotypically susceptible to carbapenems and may lead to difficulty in identifying their presence. Currently, the Infectious Diseases Society of America (IDSA) recommends the use of the novel β-lactam and β-lactamase inhibitor ceftazidime-avibactam for the treatment of bloodstream infections with OXA-CPE, with cefiderocol being an alternative (11). However, accessibility and costs are factors to consider, particularly in low- to middle-income settings. Alternative antibiotic options should also be considered.

Given the variability in phenotypic resistance that is typically demonstrated in these OXA-CPEs, we performed a retrospective review of outcomes of OXA-48-CPE and OXA-23-CPE bloodstream infections, with treatment guided by phenotypic susceptibility testing.

We retrospectively examined consecutive cases of OXA-48-CPE and OXA-23-CPE bloodstream infections managed at a single tertiary care hospital from February 2022 to July 2024. All patients were adults aged greater than 21 years of age. Blood culture isolates were identified using MALDI-TOF mass spectrometry (Bruker MALDI Biotyper, Bruker, Billerica, Massachusetts, USA). Routine susceptibility testing was performed with VITEK II (bioMérieux, Marcy-l'Étoile, France) and interpreted according to the European Committee on Antimicrobial Susceptibility Testing (EUCAST) breakpoints. OXA-48 carbapenemase production was confirmed by NG Test Carba 5 and OXA-23 by molecular methods as previously described (12).

Whole-genome sequencing (WGS) was also performed on available isolates to confirm the presence of carbapenemase genes and co-carriage of other resistance mechanisms to third-generation cephalosporins, such as ESBLs and AmpC genes. Genomic DNA was extracted from overnight plate cultures using the DNeasy Blood & Tissue Kit (Qiagen, Hilden, Germany). Sequencing was performed on an Illumina NovaSeq 6000 platform (Illumina Inc., CA, USA) to generate 150 bp paired-end reads, achieving an average depth of 120–160×. Genomes were assembled using Shovill (https://github.com/tseemann/shovill). Acquired antimicrobial resistance (AMR) determinants were screened with ABRicate (https://github.com/tseemann/abricate) and ResFinder (http://genepi.food.dtu.dk/resfinder). Sequence types (STs) were

determined using the mlst tool (https://github.com/tseemann/mlst) and MLST 2.0 (https://cge.food.dtu.dk/services/MLST/).

Clinical data were obtained from a retrospective review of the medical records. Data on the background, infection source, antimicrobial treatment strategy, and duration were tabulated. Clinical outcomes in the form of in-hospital mortality, as well as relapse of bacteremia within 1 year, were recorded. The study was approved by the institution's Domain Specific Review Board (NHG DSRB Reference number: 2024–3788). A waiver of informed consent was obtained from the above review board prior to the conduct of this retrospective study.

A total of 10 episodes of BSI from nine patients were identified, including eight with OXA-48-CPE and one with OXA-23-CPE. Only the first case of the patient with a repeated infection was included in the final analysis. The case details are summarized in Table 1. WGS data were not available for case 1. The WGS results are summarized in Table 2, with phenotypic antibiotic susceptibility results.

There were four infections caused by *Escherichia coli*, four by *Klebsiella pneumoniae*, and one by *Proteus mirabilis*. The median patient age was 63 years, with 80% ($n = 8$) male. The median Pitt bacteremia score was 0, but ranged from 0 to 9. The most common source of infection was urinary ($n = 4$, 44.4%), followed by central line-associated infections ($n = 2$, 22.2%). Four out of nine patients had been empirically initiated on meropenem, and three on piperacillin-tazobactam, before being switched to definitive antibiotics. Of the nine patients, four of the empiric antibiotics were ineffective (based on *in vitro* resistance; meropenem, $n = 1$; piperacillin-tazobactam, $n = 3$). The definitive antibiotics used include ceftriaxone ($n = 3$), ceftazidime-avibactam ($n = 2$), meropenem ($n = 2$; one as monotherapy and in combination with aztreonam), and non-β-lactam antibiotics (ciprofloxacin, $n = 1$; co-trimoxazole, $n = 2$). The total antibiotic duration ranged from 7 to 42 days. The majority of cases were treated successfully. Only one patient treated with ceftazidime-avibactam died. Of note, this patient was treated with meropenem empirically, to which his pathogen had an elevated meropenem MIC. Another patient treated with ceftazidime-avibactam had another episode of infection 10 months later and was treated with ceftazidime-avibactam again, resulting in the resolution of the infection.

All definitive antibiotics used were based on phenotypic susceptibility in the clinical microbiology report. The overall susceptibility rates to non-carbapenems are as follows: ceftazidime and cefotaxime (55.5%), amikacin (88.9%), gentamicin (66.7%), ciprofloxacin (11.1%), and co-trimoxazole (55.6%). The isolates that were resistant to 3GCs and cefepime had CTX-M-15 detected on WGS. One isolate also carried CMY-42. Two isolates with resistance to cefotaxime (but susceptible, increased exposure to cefepime and ceftazidime) carried EC-15, a chromosomally encoded group C β-lactamase (Table 2).

Currently, ceftazidime-avibactam is recommended as the preferred treatment option for OXA-48-CPE in international guidance provided by IDSA (11). In this case series, we report high rates of treatment success with treatment guided by phenotypic susceptibility testing results. This contrasts with other reports of mortality rates up to 50% (13). OXA-48 and OXA-23 vary widely in their efficiency in hydrolyzing β-lactams, with higher rates of resistance reported in OXA-23 (9, 14). Three cases (two OXA-48-like and one OXA-23) in our series were treated successfully with ceftriaxone monotherapy. This had also been reported by other authors (15). 3GCs without avibactam could thus be suitable for the treatment of these patients. In our study, the remaining patients with 3GC-susceptible isolates were also successfully treated with non-β-lactam antibiotics, with susceptibility to these agents being demonstrated.

In our series, only one *K. pneumoniae* isolate had a meropenem MIC of >8 mg/L (resistant). This patient died after receiving meropenem as empiric therapy but died despite definitive therapy with ceftazidime-avibactam. However, there was no clear association of empiric antibiotics with mortality, as three other patients received ineffective empiric antibiotics (piperacillin-tazobactam) and survived. All other isolates had variable meropenem MICs between 0.5 and 4 mg/L and were interpreted as

**TABLE 1** Summary of clinical cases, antibiotic treatment, and outcome[a]

| Case | Infection onset | Sex | Age | Organism (carbapenemase) | Co-morbidities | Pitt bacteremia score | Infection source | Empirical antibiotic choice (ST) | Definitive antibiotic choice | Antibiotic duration (days) | Outcome |
|---|---|---|---|---|---|---|---|---|---|---|---|
| 1 | 08/05/2022 | Male | 67 | Proteus mirabilis (OXA-23) | Hypertension, hyperlipidemia | 0 | Urinary source | Ceftriaxone (S) | Ceftriaxone | 14 | Survived |
| 2 | 22/08/2023 | Male | 46 | Klebsiella pneumoniae (OXA-48) | Acute myocardial infarction | 9 | Central line | Piperacillin-tazobactam (R) | Ceftriaxone | 28 | Survived |
| 3 | 22/08/2023 | Female | 50 | Escherichia coli (OXA-48) | Metastatic breast cancer, hypertension, hyperlipidemia, diabetes mellitus | 2 | Urinary source | Meropenem (S) | Ceftazidime-avibactam | 35 | Survived[b] |
| 4 | 20/10/2023 | Male | 63 | Klebsiella pneumoniae (OXA-48) | Hypertension, hyperlipidemia, diabetes mellitus, ischaemic heart disease, chronic kidney disease | 0 | Thrombophlebitis | Ceftriaxone (S) | Ceftriaxone | 42 | Survived |
| 5 | 27/10/2023 | Male | 68 | Klebsiella pneumoniae (OXA-48) | Hypertension, hyperlipidemia, diabetes mellitus, chronic kidney disease | 3 | Source unclear | Meropenem (R) | Ceftazidime-avibactam | 11 | Died |
| 6 | 28/03/2024 | Male | 57 | Escherichia coli (OXA-48) | Diffuse large B-cell lymphoma, hypertension | 0 | Neutropenic enteritis | Meropenem (S) | High-dose meropenem, Aztreonam | 10 | Survived |
| 7 | 29/03/2024 | Male | 91 | Escherichia coli (OXA-48) | Prostate cancer, hypertension, hyperlipidemia | 0 | Urinary source | Meropenem (S) | High-dose meropenem | 18 | Survived |
| 8 | 21/04/2024 | Male | 54 | Escherichia coli (OXA-48) | Acute myeloid leukemia | 0 | Central line | Piperacillin-tazobactam (R) | Ciprofloxacin | 7 | Survived |
| 9 | 30/05/2024 | Male | 71 | Klebsiella pneumoniae (OXA-48) | Hypertension, hyperlipidemia, diabetes mellitus | 0 | Urinary source | Piperacillin-tazobactam (R) | Co-trimoxazole | 10 | Survived |

[a]ST, susceptibility testing results of the empirical antibiotics; S, susceptible; R, resistant.
[b]Relapse occurred 10 months after treatment.

TABLE 2 Summary of whole-genome-sequencing data and phenotypic drug susceptibility testing results[b,c,d]

| Case | Whole-genome-sequencing data | | | | | Database accession number[a] | Carbapenems | | | Cephalosporins | | | | Non-β-lactam antibiotics | | | |
|---|---|---|---|---|---|---|---|---|---|---|---|---|---|---|---|---|---|
| | ST | OXA | ESBL | AmpC | Other bla | | ERTA | IMI | MER | FEP | CTX | CAZ | CZA | AMI | GEN | CIP | SXT |
| 1 | N/A | | | | | | ≤0.5 (S) | 8 (R) | 1 (S) | ≤1.0 (S) | ≤1.0 (S) | ≤1.0 (S) | N/A | ≤2.0 (S) | ≤1.0 (S) | 2.0 (R) | ≥16 (R) |
| 2 | 76 | OXA-181 | | | LAP-2, SHV-187 | SAMN44581316 | 4 (R) | 2 (S) | 2 (S) | ≤1.0 (S) | ≤1.0 (S) | ≤1.0 (S) | ≤0.5 (S) | ≤2.0 (S) | ≤1.0 (S) | 1 (R) | ≤1 (S) |
| 3 | 410 | OXA-484 | CTX-M-15 | CMY-42 | EC-15 | SAMN44581338 | 4 (R) | 2 (S) | 1 (S) | ≥64.0 (R) | ≥64.0 (R) | ≥64.0 (R) | 2 (S) | 8 (S) | ≤1.0 (S) | ≥4.0 (R) | ≤1 (S) |
| 4 | 611 | OXA-181 | | | SHV-27 | SAMN44581350 | 4 (R) | 2 (S) | 1 (S) | ≤1.0 (S) | ≤1.0 (S) | ≤1.0 (S) | N/A | ≤2.0 (S) | ≤1.0 (S) | 1 (R) | ≤1 (S) |
| 5 | 231 | OXA-232 | CTX-M-15 | | SHV-212, TEM-1 | SAMN44581310 | ≥8.0 (R) | 2 (S) | ≥16 (R) | ≥64.0 (R) | ≥64.0 (R) | ≥64.0 (R) | 1 (S) | ≥64.0 (R) | ≥16.0 (R) | ≥4.0 (R) | ≥16 (R) |
| 6 | 410 | OXA-484 | | | EC-15 | SAMN44581351 | ≥8.0 (R) | 2 (S) | 2 (S) | 2 (I) | 8 (R) | 4 (I) | ≤0.5 (S) | ≤2.0 (S) | ≥16.0 (R) | ≥4.0 (R) | ≥16 (R) |
| 7 | 410 | OXA-484 | | | EC-15, TEM-1 | SAMN44581352 | 2 (R) | 1 (S) | 1 (S) | 2 (I) | 8 (R) | 4 (I) | 1 (S) | ≤2.0 (S) | ≥16.0 (R) | ≥4.0 (R) | ≥16 (R) |
| 8 | 442 | OXA-244 | | | EC-18 | SAMN44581311 | 4 (R) | 0.5 (I) | 2 (S) | ≤1.0 (S) | ≤1.0 (S) | ≤1.0 (S) | ≤0.5 (S) | ≤2.0 (S) | ≤1.0 (S) | ≤0.25 (S) | ≤1 (S) |
| 9 | – | OXA-181 | | | SHV-187 | SAMN44581316 | 2 (R) | 1 (S) | 0.5 (S) | ≤1.0 (S) | ≤1.0 (S) | ≤1.0 (S) | N/A | ≤2.0 (S) | ≤1.0 (S) | 1 (R) | ≤1 (S) |

[a]Data from this study are publicly available in the NCBI BioProject database under accession number PRJNA1182112.
[b]Phenotypic drug susceptibility testing expressed as minimum-inhibitory concentration in mg/L, with interpretations in parentheses.
[c]ST, sequence type; ESBL, extended-spectrum β-lactamase; bla, β-lactam; ERTA, ertapenem; IMI, imipenem; MER, meropenem; FEP, cefepime; CTX, cefotaxime; CAZ, ceftazidime; CZA, ceftazidime-avibactam; AMI, amikacin; GEN, gentamicin; CIP, ciprofloxacin; SXT, trimethoprim-sulfamethoxazole; S, susceptible, standard dose; I, susceptible, increased exposure; R, resistant; N/A, Not available.
[d]Shaded cells indicate phenotypic resistance.

susceptible. For definitive antibiotic therapy, other β-lactams used include meropenem as monotherapy ($n = 1$) and in combination with aztreonam ($n = 1$). While there have been reports of treatment success of OXA-48-like CPE with carbapenems where the MIC is low, this is typically not recommended if there are alternative effective antimicrobials (11, 16). Aztreonam was included in combination for an isolate that was susceptible to increased exposure. The only documented treatment failure (death) occurred in a patient treated with ceftazidime-avibactam. This may have been a reflection of the underlying host being more critically ill, given that the onset of the bacteremia was in the intensive care unit, with a Pitt bacteremia score of 3.

Routine susceptibility testing for cefotaxime, ceftazidime, and cefepime was performed using Vitek II. The phenotypic results were concordant across all three antibiotics for all except for two isolates. These two isolates were susceptible to increased exposure to cefotaxime and cefepime, despite resistance to ceftazidime. Although EUCAST recommends that interpretations be reported as tested, in practice, our laboratory reports all as resistant on the basis that resistance to any could be an indicator of the presence of an extended-spectrum β-lactamase. Although the numbers are small, this practice is supported by the correlation of the phenotype and genotype based on the WGS results (Table 2).

In summary, we report the potential for third-generation cephalosporins as a viable treatment option for OXA-48 and OXA-23-CPE bloodstream infections, when phenotypic susceptibility is demonstrated, although our low numbers limit generalizability. Despite the limitations and exploratory nature of our findings, it highlights the potential of 3GCs as a safe treatment option for some OXA-48 and OXA-23-CPE infections, which could be further examined in larger studies.

## ACKNOWLEDGMENTS

Whole-genome sequencing was supported by the National University Health System (NUHS) seed grant fund to Jeanette Teo (Grant number: NUHSRO/2023/036/RO5+6/Seed-Mar/01).

M.C.Y.K., J.N.N., N.C.J.H., and C.K.L. contributed to the conception, data collection, data analysis, and original draft of the manuscript. J.T. performed whole-genome sequencing and bioinformatics analysis of isolates.

## AUTHOR AFFILIATIONS

[1]Division of Infectious Diseases, Department of Medicine, National University Health System, Singapore, Singapore
[2]Department of Laboratory Medicine, National University Hospital, Singapore, Singapore

## AUTHOR ORCIDs

Jinghao Nicholas Ngiam ⓘ http://orcid.org/0000-0002-3339-7281
Matthew Chung Yi Koh ⓘ http://orcid.org/0009-0002-7374-8207
Jeanette Teo ⓘ http://orcid.org/0000-0002-9321-3341
Ka Lip Chew ⓘ http://orcid.org/0000-0002-4654-6434

## AUTHOR CONTRIBUTIONS

Jinghao Nicholas Ngiam, Conceptualization, Data curation, Formal analysis, Writing – original draft, Writing – review and editing | Matthew Chung Yi Koh, Conceptualization, Data curation, Formal analysis, Writing – original draft, Writing – review and editing | Nicholas Jian Hao Chan, Conceptualization, Data curation, Formal analysis, Writing – review and editing | Jeanette Teo, Conceptualization, Data curation, Formal analysis, Writing – review and editing | Ka Lip Chew, Conceptualization, Data curation, Formal analysis, Writing – review and editing

## DATA AVAILABILITY

Data from this study are publicly available in the NCBI BioProject database under accession number PRJNA1182112.

## ETHICS APPROVAL

This study was approved by the hospital's Institutional Review Board (National Healthcare Group (NHG) Domain Specific Review Board (DSRB) reference number: 2024–3788

## ADDITIONAL FILES

The following material is available online.

### Open Peer Review

**PEER REVIEW HISTORY (review-history.pdf).** An accounting of the reviewer comments and feedback.

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
