## [Reviewer comments · Microbiology Spectrum]

Microbiology Spectrum

Clinical outcomes of Phenotype-Guided Treatment in Group D Carbapenemase-Producing Enterobacterales Bacteremia

Jinghao Nicholas Ngiam, Matthew Koh, Nicholas Chan, Jeanette TEO, and Chew Ka Lip

Corresponding Author(s): Jinghao Nicholas Ngiam, National University Health System

Review Timeline:

Submission Date:	January 26, 2025
Editorial Decision:	February 25, 2025
Revision Received:	February 27, 2025
Accepted:	March 18, 2025

Editor: Po-Yu Liu

Reviewer(s): Disclosure of reviewer identity is with reference to reviewer comments included in decision letter(s). The following individuals involved in review of your submission have agreed to reveal their identity: Yehuda Carmeli (Reviewer #2)

Transaction Report:

DOI: <https://doi.org/10.1128/spectrum.00273-25>

Re: Spectrum00273-25 (Clinical outcomes of OXA-carbapenemase-producing Enterobacterales bloodstream infections with ceftriaxone treatment)

Dear Dr. Jinghao Nicholas Ngiam:

Thank you for the privilege of reviewing your work. They recommend significant revisions to your manuscript to address reviewer feedback. Key revisions include mandatory data deposition with accession numbers, clear differentiation between OXA-48-like and OXA-23, specification of cephalosporin names, a refocused or expanded scope regarding ceftriaxone treatment, detailed description of empiric therapy and bioinformatics methodology, addressing limitations of VITEK II for MIC testing, and enhanced case descriptions. To better reflect the focus and scope of your study, especially given the reviewer comments and emphasis on ceftriaxone in a subset of cases, I suggest revising your title to accurately reflect the study's scope and address the concern of overstatement. Please address these points comprehensively in your revision and provide a detailed response to reviewers.

Revision Guidelines

Sincerely,
Po-Yu Liu
Editor
Microbiology Spectrum

Reviewer #1 (Comments for the Author):

The authors present nine cases of infection by Enterobacterales producing OXA-type carbapenemase at a single institution in Singapore. Five of these isolates did not harbor other broad-spectrum beta-lactamase genes and were susceptible to third-generation cephalosporins. Three of these cases were actually treated with ceftriaxone and achieved cure. Although the number of cases is small, this report is of a certain value in exploring treatment options for multidrug-resistant bacterial infections. Some suggestions for modification are presented below.

Major comments

- In this manuscript, OXA-48-like and OXA-23 are described collectively, but the epidemiological position of the two is quite different. OXA-48-like is one of the major carbapenemases produced by carbapenemase-producing Enterobacterales, whereas OXA-23 is mainly produced by *A. baumannii*. The enzymatic characteristics of OXA-23 and the antimicrobial susceptibility and epidemiology of OXA-23-producing Enterobacterales should be described in the Introduction and Discussion separately from the OXA-48-like-producing Enterobacterales. In fact, there was only one case of OXA-23-producing Enterobacterales in the present study. Although it would have been preferable to exclude this case because whole genome sequencing analysis was not performed on this strain only, I avoided suggesting this because it is one of only three ceftriaxone-treated cases.
- Whole genome sequencing analysis data should be registered in public databases and accession numbers should be provided in the manuscript.

Major comments

- Lines 149-150: "cephalosporins" should be described by specific antimicrobial names.

Reviewer #2 (Comments for the Author):

The study refers to ceftriaxone treatment of OXA producing enterobacterales, however among the 9 included patients only 3 have been treated with ceftriaxone. Suggest to either focus on describing the 3 patients or change the focus of the paper to address all treatments. Information on treatment of 3 patients with ceftriaxone may be addition to scant literature, however, it is difficult to draw any conclusions from such a small number.

The empiric therapy before targeted therapy is not described and may have important effect on outcome. The MIC of the isolates was determined using VITEK II, which is not a gold standard method to determine MICs. In this small study where MICs are of great importance, a gold standard method should be used. The bioinformatics methodology is not described.

Manuscript Ref. No.: Spectrum00273-25

Title: Clinical outcomes of OXA-carbapenemase-producing Enterobacterales bloodstream infections with ceftriaxone treatment

Updated title: Clinical outcomes of Phenotype-Guided Treatment in Group D Carbapenemase-Producing Enterobacterales Bacteremia

Microbiology Spectrum

We thank the Editor for the opportunity to revise our manuscript and the Reviewers for the constructive comments. We have made changes to improve our paper in order to address the points raised by the Reviewers.

We have uploaded two version of the manuscript, one with marked changes in red font, and the other without marked changes.

In the sections below, each of the points raised is identified and addressed with changes in the revised manuscript.

Editor's comments:

Thank you for the privilege of reviewing your work. They recommend significant revisions to your manuscript to address reviewer feedback. Key revisions include mandatory data deposition with accession numbers, clear differentiation between OXA-48-like and OXA-23, specification of cephalosporin names, a refocused or expanded scope regarding ceftriaxone treatment, detailed description of empiric therapy and bioinformatics methodology, addressing limitations of VITEK II for MIC testing, and enhanced case descriptions. To better reflect the focus and scope of your study, especially given the reviewer comments and emphasis on ceftriaxone in a subset of cases, I suggest revising your title to accurately reflect the study's scope and address the concern of overstatement. Please address these points comprehensively in your revision and provide a detailed response to reviewers.

We have no added details with regards to empiric therapy (Table 1) and associated resistance results, bioinformatics, and more discussion on the use of routine susceptibility testing results, in this case Vitek II.

We have also revised the title to broaden the scope, that is to say that with regard to OXA-CPE, we should consider whether more weight should be put on the phenotype (i.e. susceptibility) as compared to genotype:

“Clinical outcomes of Phenotype-Guided Treatment in Group D Carbapenemase-Producing Enterobacterales Bacteremia”

Reviewer #1 (Comments for the Author):

The authors present nine cases of infection by Enterobacterales producing OXA-type carbapenemase at a single institution in Singapore. Five of these isolates did not harbor other broad-spectrum beta-lactamase genes and were susceptible to third-generation

cephalosporins. Three of these cases were actually treated with ceftriaxone and achieved cure.

Although the number of cases is small, this report is of a certain value in exploring treatment options for multidrug-resistant bacterial infections. Some suggestions for modification are presented below.

Please refer to response to reviewer 2 with regards to the small number of cases.

Major comments

- In this manuscript, OXA-48-like and OXA-23 are described collectively, but the epidemiological position of the two is quite different. OXA-48-like is one of the major carbapenemases produced by carbapenemase-producing Enterobacterales, whereas OXA-23 is mainly produced by *A. baumannii*. The enzymatic characteristics of OXA-23 and the antimicrobial susceptibility and epidemiology of OXA-23-producing Enterobacterales should be described in the Introduction and Discussion separately from the OXA-48-like-producing Enterobacterales. In fact, there was only one case of OXA-23-producing Enterobacterales in the present study. Although it would have been preferable to exclude this case because whole genome sequencing analysis was not performed on this strain only, I avoided suggesting this because it is one of only three ceftriaxone-treated cases.

We thank the Reviewer for pointing this out. We have re-written our introduction and discussion to describe OXA-48 and OXA-23 separately. We hope this adds to the clarity of our manuscript (Page 6, 8-9)

- Whole genome sequencing analysis data should be registered in public databases and accession numbers should be provided in the manuscript.

We have now included the data availability statement. As the sequencing data was generated as part of sequencing a larger collection (non-blood) CPE isolates, we have also made note of the accession numbers in Table 2 for ease of reference for interested readers.

Data availability Statement: Data from this study are publicly available in the NCBI BioProject database under accession number PRJNA1182112. Individual accession numbers of each isolate have also been added to the table 2.

Major comments

- Lines 149-150: "cephalosporins" should be described by specific antimicrobial names.

We have replaced this with "ceftazidime and cefotaxime" for greater clarity (Page 7).

Reviewer #2 (Comments for the Author):

The study refers to ceftriaxone treatment of OXA producing enterobacteriales, however among the 9 included patients only 3 have been treated with ceftriaxone. Suggest to either focus on describing the 3 patients or change the focus of the paper to address all treatments. Information on treatment of 3 patients with ceftriaxone may be addition to scant literature, however, it is difficult to draw any conclusions from such a small number.

We thank the Reviewer for these comments. We acknowledge the small sample size as a limitation, making it difficult to make generalisations and draw conclusions. However, as pointed out, there is a paucity of literature describing treatment outcomes in OXA-48-CPE that remain susceptible to cephalosporins. To our knowledge, there is only one article (citation 16, PMID: 30825700).

This may be due to several reasons; in conversation with other colleagues, some labs may not report 3rd-generation-cephalosporins as susceptible where there is OXA-48 detected. Many clinicians may not use 3rd-generation-cephalosporins as monotherapy due to there being limited reports.

We highlight the exploratory nature and limitations of our work in the summarizing paragraph in the discussion

The empiric therapy before targeted therapy is not described and may have important effect on outcome.

We have added in the empiric antibiotic choice in the table 1, and indicated whether the antibiotic was susceptible or resistant in parentheses . This was not shown to be associated with adverse outcomes. Four patients were empirically treated with drugs for which there was *in vitro* resistance, of these, one patient with meropenem-resistance died.

The MIC of the isolates was determined using VITEK II. which is not a gold standard method to determine MICs. In this small study where MICs are of great importance, a gold standard method should be used.

Unfortunately, our clinical laboratory does not have reference broth microdilution testing available. Instead, testing was performed for three cephalosporins, as an indicator of whether the isolates are likely or not to contain a ESBL/AmpC. The phenotypic results also corroborated strongly with our WGS findings, again supporting validating of the Vitek results in separating true susceptible and resistant results.

One of the safeguards that is used by the clinical laboratory is reporting the 3rd-generation cephalosporins (and 4th-generation for non-chromosomal AmpC Enterobacteriales) as resistant if there is an increase in MIC to any of tested cephalosporins. Although this differs from EUCAST recommendations, we think that this

is a safe approach in deciding on treatment with 3rd-generation-cephalosporins for our context. We have thus added a paragraph in the discussion:

“Routine susceptibility testing for cefotaxime, ceftazidime, and cefepime was performed using Vitek II. The phenotypic results were concordant across all three antibiotics for all except for two isolates. These two isolates were susceptible, increased exposure to cefotaxime and cefepime despite resistance to ceftazidime. Although EUCAST recommends that interpretations are reported as tested, in practice, our laboratory report all as resistant on the basis that resistance to any could be an indicator of the presence of an extended-spectrum- β -lactamase. Although the numbers are small, this practice is supported by the correlation of the phenotype and genotype based on the WGS results (Table 2).”

The bioinformatics methodology is not described.

We have described the bioinformatics method in greater detail as well.

Once again, we thank the Editor and Reviewers for the helpful and constructive comments. We hope the paper is now suitable for publication in the Journal.

Warmest Regards,

Dr Nicholas Ngiam, Dr Matthew Koh and Dr Chew Ka Lip

Re: Spectrum00273-25R1 (Clinical outcomes of Phenotype-Guided Treatment in Group D Carbapenemase-Producing Enterobacterales Bacteremia)

Dear Dr. Jinghao Nicholas Ngiam:

Your manuscript has been accepted, and I am forwarding it to the ASM production staff for publication. Your paper will first be checked to make sure all elements meet the technical requirements. ASM staff will contact you if anything needs to be revised before copyediting and production can begin. Otherwise, you will be notified when your proofs are ready to be viewed.

Sincerely,
Po-Yu Liu
Editor
Microbiology Spectrum

Reviewer #1 (Comments for the Author):

The authors revised the manuscript appropriately in accordance with the reviewers' suggestions.

Whether Enterobacterales carrying the carbapenemase gene or other beta-lactamase genes can be successfully treated based on phenotype is a crucial aspect of antimicrobial stewardship. This article effectively explores this direction, successfully conveying its objectives and core message.

However, achieving this research goal may require a larger study sample to minimize bias. I believe the current data volume is insufficient to fully support the conclusion. Extending the study period to collect more cases is recommended for stronger validation.